# Impact of Green Chitosan Nanoparticles Fabricated from Shrimp Processing Waste as a Source of Nano Nitrogen Fertilizers on the Yield Quantity and Quality of Wheat (*Triticum aestivum* L.) Cultivars

**DOI:** 10.3390/molecules27175640

**Published:** 2022-09-01

**Authors:** Ahmed M. Saad, Aya Yaseen Mahmood Alabdali, Mohamed Ebaid, Eslam Salama, Mohamed T. El-Saadony, Samy Selim, Fatmah A. Safhi, Salha M. ALshamrani, Hanan Abdalla, Ayman H. A. Mahdi, Fathy M. A. El-Saadony

**Affiliations:** 1Agronomy Department, Faculty of Agriculture, Benha University, Benha 13511, Egypt; 2Department of Pharmacy, Faculty of Pharmacy, The University of Mashreq, Baghdad 10023, Iraq; 3Plant Production Department, Arid Lands Cultivation Research Institute (ALCRI), City of Scientific Research and Technological Applications (SRTA-City), New Borg El-Arab City, Alexandria 21934, Egypt; 4Environment and Natural Materials Research Institute (ENMRI), City of Scientific Research and Technological Applications (SRTA-City), New Borg El-Arab City, Alexandria 21934, Egypt; 5Department of Agricultural Microbiology, Faculty of Agriculture, Zagazig University, Zagazig 44511, Egypt; 6Department of Clinical Laboratory Sciences, College of Applied Medical Sciences, Jouf University, Sakaka 72388, Saudi Arabia; 7Department of Biology, College of Science, Princess Nourah bint Abdulrahman University, P.O. Box 84428, Riyadh 11671, Saudi Arabia; 8Department of Biology, College of Science, University of Jeddah, Jeddah 23341, Saudi Arabia; 9Botany and Microbiology Department, Faculty of Science, Zagazig University, Zagazig 44511, Egypt; 10Agronomy Department, Faculty of Agriculture, Beni-Suef University, Beni-Suef 62521, Egypt; 11Agricultural Botany Department, Faculty of Agriculture, Zagazig University, Zagazig 44511, Egypt

**Keywords:** shrimp waste, chitosan nanoparticles, nano-nitrogen, mineral nitrogen, fertilizers, wheat quality

## Abstract

Waste from crustaceans has adverse effects on the environment. In this respect, shrimp waste was valorized for producing chitosan nanoparticles as a source for eco-friendly nano-nitrogen fertilizer. The application of nano-nitrogen fertilizers is a valuable alternative approach in agriculture due to its potential for reducing the application of mineral nitrogen fertilizers and increasing yield quality and quantity, thereby helping to reduce the worldwide food shortage. Chitosan nanoparticles were foliar sprayed at three volumes (0, 7, and 14 L/ha) and compared with mineral nitrogen fertilizer (M-N) sprayed at three volumes (0, 120, and 240 kg N/ha) and their combination on two wheat cultivars (Misr-1 and Gemaiza-11) during two consecutive seasons (2019/2020 and 2020/2021) in order to evaluate the agronomic response. The synthesized chitosan nanoparticles displayed characteristic bands of both Nan-N and urea/chitosan from 500–4000 cm^−1^. They are stable and have a huge surface area of 73.21 m^2^ g^−1^. The results revealed significant differences among wheat cultivars, fertilization applications, individual or combined, and their interactions for yield-contributing traits. Foliar application of nano-nitrogen fertilizer at 14 L/ha combined with mineral fertilizer at 240 kg/ha significantly increased total chlorophyll content by 41 and 31% compared to control; concerning plant height, the two cultivars recorded the tallest plants (86.2 and 86.5 cm) compared to control. On the other hand, the heaviest 1000-grain weight (55.8 and 57.4 g) was recorded with treatment of 120 kg Mn-N and 14 L Nan-N/ha compared to the control (47.6 and 45.5 g). The Misr-1 cultivar achieved the highest values for grain yield and nitrogen (1.30 and 1.91 mg/L) and potassium (9.87 and 9.81 mg/L) in the two studied seasons when foliarly sprayed with the combination of 120 kg Mn-N/ha + 14 L Nan-N/ha compared to the Gemaiza-11 cultivar. It can be concluded that Misr-1 exhibited higher levels of total chlorophyll content, spike length, 100-grain weight, grain yield in kg/ha, and nitrogen and potassium. However, Gemaiza-11 displayed higher biomass and straw yield values, plant height, and sodium concentration values. It could be economically recommended to use the application of 120 kg Mn-N/ha + 14 L Nan-N/ha on the Misr-1 cultivar to achieve the highest crop yield.

## 1. Introduction

Crustacean waste causes several environmental, agricultural, and soil issues, and serves as a vector for the *Aphanomyces staci* fungus. Utilizing shrimp processing waste as a source of chitosan has the potential to turn risks into beneficial uses. There are other methods for preparing chitosan; however, this one provides the best physical and chemical characterization in size, although the surface shape remains undetermined. Chitosan is gaining popularity because it satisfies environmental criteria, such as being an eco-friendly chemical that facilitates the effective use of reagents while minimizing possible waste [1]. Chitosan is a natural polymer derived from the deacetylation of chitin. A positive effect of chitosan has been observed on the growth of various plants’ roots, shoots, and leaves, including gerbera [2,3], and several other plants [4]. 

Wheat (*Triticum aestivum* L.) is a major cereal crop in Egypt. Its grains supply about 70% of calories and 80% of the protein in the human diet [5]. In 2020, the wheat cultivated area in Egypt was 1.37 million hectares, producing 9.5 million tons/year [6]. This amount covers less than 55% of the local consumer demand, a deficit estimated at 45% of wheat grain [7]. The main objective of the Egyptian Government is to reduce the gap between production and consumption by increasing the productivity and quality of wheat cultivars using appropriate mineral and nano-fertilizer nitrogen [8].

Wheat growers select cultivars depending on their production capacity, maturity, winter hardiness, straw strength, spike length, plant height, lodging resistance, seed size, seed weight, bread quality, and other improved traits to be taken into consideration [9]. They also pay special attention to new promising varieties being developed, distinguished by early ripening and high production capacity. Selecting a wheat cultivar at the right time best ensures optimal flowering and, consequently, maximum yield [10].

Mineral nitrogen (M-N) is the first and foremost nutrient required for plants. Nitrogen plays an efficacious role in achieving a high yield of grains and protein when an adequate balance of nitrogen nutrients is achieved, as both excess and deficiency of N have adverse effects on crop growth and development [11]. Increasing harmful forms of nitrogen (nitrate, ammonia, and nitrogen oxides) in the soil results in environmental damage, such as air pollution and the promotion of global climate change, nutrient enrichment, and soil acidification [12]. Therefore, nitrogen use efficiency (NUE) should be evaluated in order to maximize the benefit of nitrogen fertilization and emphasize the N crop response. Optimal nitrogen management is vital for maximizing productivity and minimizing pollution [12].

The application of nanotechnology in agriculture has been increasing over recent years and constitutes a valuable tool to achieve the goal of sustainable food production worldwide [13,14,15]. The properties and possibilities of nanotechnology, which are of great interest in the agricultural revolution, are high reactivity, enhanced bioavailability and bioactivity, adherence effects, and surface effects of nanoparticles [16]. 

These unique properties are due to the tiny molecular size and the modified interactions between molecules. Nano-fertilizers have been studied to increase nutrient efficiency and improve plant nutrition compared to traditional fertilizers. A nano-fertilizer is any nanoparticle product used to improve nutrient efficiency [17]. Chitosan nanoparticles may be considered a Nan-N that efficiently supplies plants with chemicals and nutrients [18]. Therefore, applying nano-fertilizers alone or in combination with mineral fertilizers reduces environmental pollution owing to significantly fewer losses and a higher nitrogen absorption rate.

Nano-fertilizers enhance growth parameters, i.e., plant height, leaf area, number of leaves per plant, dry matter production, chlorophyll production, and the photosynthesis rate, which result in more production and translocation of photosynthesis to different parts of the plant compared to mineral fertilizers, as reported by Al-Juthery et al., [16], Singh [19]. Applying nano-fertilizers with low doses of mineral N fertilizers can boost the productivity of cereal crops [20]. Therefore, this study aims to valorize shrimp waste by producing chitosan nanoparticles as a source of safe nitrogen fertilizer. Additionally, it aims to evaluate the yield potential and its components for two wheat cultivars, Misr-1 and Gemeiza-11, using a combination of chitosan nanoparticles (Nan-N) and mineral nitrogen (M-N) fertilization levels.

## 2. Materials and Methods

### 2.1. Isolation of Chitosan from Shrimp Waste

Shrimp waste (2 kg) was acquired from a local fish market and rinsed with tap water many times to eliminate impurities. The powder was then dried in an oven at 80 °C, powdered, and stirred with 50 mL of 10% hydrochloric acid (HCl) (1:5, *w*/*v*) for 12 h at room temperature; the pH was adjusted to 7 using a pH meter. The residues were separated, 5% sodium hydroxide (NaOH) was added (1:10, *w*/*v*) to remove the protein, and the residues were washed and dried at 80 °C. The dry materials were agitated for 12 h at 70 °C in acetone while a reflecting condenser returned the evaporated acetone to the extraction vessel. This chitin was then dried in an oven at 80 degrees Celsius. The obtained chitin was deacetylated with 50 percent NaOH under stirring at 200 rpm, 115 °C for 24 h under a reflective condenser to maintain a constant mixture volume; the chitosan was obtained, washed, pH adjusted to 7, dried at 70 °C, and then purified by dissolving in acetic acid followed by precipitating by wise adding NaOH (1M). It was then neutralized by sterilized distilled water and freeze-dried [1]. 

### 2.2. Biological Preparation of Chitosan Nanoparticles

*Penicillium oxalicum* was obtained from the Department of Agricultural Microbiology, Faculty of Agriculture, Zagazig University, Zagazig, Egypt, and grown in Czepak Dox broth medium for three days. The *Penicillium oxalicum* extracellular proteins were precipitated with 80 percent (*w*/*v*) ammonium sulfate saturation. As mentioned before, 5 mg of precipitated proteins were passed over a 30 × 2.5 cm CM-cellulose column that had been pre-saturated [21]. The unbound proteins eluted from the column were collected and analyzed for protein concentration to prepare chitosan nanoparticles. A total of 0.5 % (*w*/*v*) shrimp chitosan was dissolved in 1 % (*v*/*v*) acetic acid, and the pH was adjusted to 4.8. The 180 g/mL ionic proteins eluted from the column’s void volume (6 mL) was added to the chitosan solution (15 mL) with magnetic stirring for 30 minu and maintained at room temperature overnight. The colloidal solution was centrifuged at 10,000× *g* for 10 min following incubation. The precipitate was washed twice to eliminate unreacted material and freeze-dried. The obtained chitosan nanoparticles were used as a source of nitrogen fertilization. 

### 2.3. Characterization of Chitosan Nanoparticles 

The fabricated chitosan nanoparticles were characterized using an infrared absorption spectrum (BRUKER, Germany) to detect the bioactive groups in the synthesized fertilizer [22]. X-ray diffraction (XRD) of the chitosan nanoparticles was studied by Shimadzu-7000 in order to explain the crystallinity of the fabricated nano-fertilizer [23]. Scanning electron microscopy (SEM) (JEOL JSM-6010LV) and transmission electron microscopy (TEM) (JEOL JEM-2100) were utilized for the investigation of the morphological structure of nano-fertilizer [24]. The energy dispersive X-ray (EDX) technique was used to analyze the elemental profile of chitosan nanoparticles. The thermal stability of chitosan nanoparticles was calculated by TGA-50 Shimadzu. The fertilizer weight loss was recorded against a temperature range from 28 to 800 °C under nitrogen. The surface area and pore size of the synthesized nano-fertilizer were estimated by degassing the prepared fertilizer under a vacuum at 25 °C for 12 h using Brunauer-Emmett-Teller (BET) (Beckman Coulter SA3100, Brea, CA, USA).

### 2.4. Plant Material, Growing Conditions

Wheat cultivars Gemaiza-11 and Misr-1 were provided by the Wheat Department, Agriculture Research Center, Ministry of Agriculture at Giza, Egypt. The seed rates of each cultivar were used as a recommendation for wheat yield production packages. Two field experiments were conducted at the Experimental Research Station, Faculty of Agriculture, Benha University, Benha, Egypt during two consecutive seasons (2019/2020 and 2020/2021) to evaluate the response of two wheat cultivars (Misr-1 and Gemaiza-11) under foliar spray at three volumes of nano-nitrogen (Nan-N) fertilizer chitosan nanoparticles (0, 7, and 14 L/ha), three volumes of mineral nitrogen (Mn-N) fertilizer (0, 120 and 240 kg N/ha), and their combination. Grains were sown on 24 November in the two growing seasons. The experimental unit was 10.5 m^2^ in area (3 × 3.5 m). The soil of the experimental site was analyzed during each of the two seasons according to Dahnke and Whitney [25], and the results were recorded in Appendix A.

The chitosan nanoparticles (CSN) were prepared by polymerization of methacrylic acid (MAA) in a two-step process according to de Moura, et al. [26] and Corradini, et al. [27]. The incorporation of N fertilizer in CSN was achieved by dissolving suitable amounts of N in CSN (600 ppm). Three CSN (Nan-N) levels of 0, 7, and 14 L/ha were prepared from a stock solution of 600 ppm and loaded in liquid urea/chitosan. Plants were foliarly sprayed twice before both the first and the second irrigations (30 and 70 days post-sowing, DAS). 

Three volumes of nitrogen (Mn-N)—0, 120, and 240 kg N/ha—loaded in 46.5% liquid urea N were used. Each level was divided into two equal doses; the first was applied before the first irrigation, and the second dose was added before the second one. 

The experiment was laid out in a split-plot design in randomized complete blocks (RCBD) with three replicates, where cultivars were randomly distributed in the main plots, while nano and mineral nitrogen fertilizers were randomly located in the split-plots. 

### 2.5. Studied Parameters

#### 2.5.1. Yield and Its Attributes

The fresh leaves were collected and extracted in 80% acetone. The total chlorophyll content was measured using a chlorophyll meter (SPDS) Model SPAD-402, CDI, 3963 Walnut St., Denver, CO, 80205, according to Mielke, et al. [28], plant height (cm), number of tillers/m^2^, spike length (cm), number of spikelets/spike, spike weight (g), 1000-grain weight (g), grain yield (kg/ha), straw yield (kg/ha), and biological yield (kg/ha).

#### 2.5.2. Determination of Nitrogen, Sodium, and Potassium Contents

The total nitrogen in grain wheat was determined using the ammonia/nitrogen distillation unit (Kjeldahl distiller Nitrogen–Protein DNP, RAYPA, Barcelona, Spain). The sodium and potassium contents in the wheat grain were analyzed via analytical atomic absorption spectroscopy (Model AAS GPC A932, Version 1.1, Perkin-Elmer, UK). Dried leaves were digested with concentrated nitric acid at a temperature of 100 °C until a transparent solution was reached. The obtained solution was diluted to a known volume with distilled water. 

### 2.6. Statistical Analysis 

The R statistical software version 4.1.1 was applied to analyze the data. Analysis of variance was performed for both seasons using a split-plot design with the cultivars in the main plots and the applied fertilizers in the split-plots according to Steel [29]. The differences among the evaluated cultivars, applied fertilizers, and their interaction were separated by the least-significant difference at *p* ≤ 0.05. 

## 3. Results 

### 3.1. Characterization of Chitosan Nanoparticles

The morphological structure of chitosan nanoparticles was detected using SEM and TEM, as presented in Figure 1A,B. The presence of large particles in SEM attributed to the agglomeration and combination of the organic matrices. Both SEM and TEM pictures clarify the nanomorphology of the fabricated nano-fertilizer with acceptable uniformity of size, which is very useful in fertilization applications and varies from the previously described fertilizers that were synthesized with a large size distribution in the micro-scale [30]. The decrease in the size of the prepared fertilizer may be due to the modifications in the synthesis conditions, which improved and reduced the formation of nanoparticles compared to the previously reported conditions in the literature [31]. 

FT-IR spectra of the synthesized chitosan nanoparticles displays the characteristic bands of both Nan-N and urea/chitosan from 500–4000 cm^−1^. The FTIR spectra shows a broad peak around 3400–3500 cm^−1^ attributed to the stretching vibration of amino and hydroxyl groups of the fabricated matrices [32,33]. The antisymmetric bands around 1650 cm^−1^ can be assigned to the carbonyl of urea and chitosan, although the symmetric carbonyl stretching peak was detected at 1454 cm^−1^, as shown in Figure 1C. Furthermore, chitosan exhibited a characteristic peak in the CH_3_ group and CH_3_-O at 1153 cm^−1^. The identified band at 1393 cm^−1^ was due to the C-C stretching [32,34,35]. However, several bands clarify the incorporated micronutrients in nano-fertilizer localized at the wavelength range 500–1150 cm^−1^ [36]. 

The crystalline degree of chitosan nanoparticles was investigated via XRD, as shown in Figure 1D. The XRD pattern displays diverse sharpness peaks of urea at 21.6°, 28.3°, and 35.0° [37]. Also, it shows a peak at 2θ of 16.1°, which is attributed to chitosan. These findings are characteristic of the most distinctive signals of the nano fertilizers’ nitrogen with a high crystallinity without any impurities [38,39]. The elemental analysis of chitosan nanoparticles was carried out via the energy dispersive X-ray (EDX) technique to confirm the homogenous distribution of each of the composite components without any impurities, as shown in Figure 1E. The electron images of chitosan nanoparticles were observed in Appendix A.

The thermal stability of the fabricated chitosan nanoparticles was tested under nitrogen gas, and the thermogram was presented in Appendix A with many weight-loss phases. The first phase was about 30% at 220 °C, which was related to the degradation of water molecules and atmospheric gases that were suspended with the pores of the nano-fertilizer. The second phase was about 23% and appeared at the range of 220 °C to 1000 °C, which may be due to the decomposition of the nano-fertilizer components [40,41,42]. These data demonstrated the high thermal stability of chitosan nanoparticles. 

The surface area profile of the fabricated chitosan nanoparticles was detected by Brunauer−Emmett−Teller (BET), as shown in Figure 1F. The mean pore diameter was about 9.97 nm, the surface area of the synthesized nano-fertilizer was recorded 73.21 m^2^ g^−1^, and the pore volume was 0.018 cm^3^ g^−1^. The identified high surface area of the synthesized fertilizer is helpful for applying this material in agricultural applications [31].

### 3.2. Yield Properties 

#### 3.2.1. Total Chlorophyll Content

Data presented in (Table 1) revealed significant differences between the two cultivars in terms of total chlorophyll content in both seasons. The Misr-1 cultivar recorded the highest values of total chlorophyll content (44.2 and 44.3 mg/g). Meanwhile, Gemaiza-11 recorded the lowest values (43.8 and 43.5 mg/g) in the two studied seasons, respectively. Furthermore, the total chlorophyll content was significantly affected by applying different nano- and mineral nitrogen fertilizer levels in the two growing seasons (Table 1). The combination of 240 kg Mn-N/ha with 14 L Nan-N/ha significantly increased total chlorophyll content by 41 and 31% compared to the untreated control in the two seasons, respectively. Otherwise, the minimum total chlorophyll content (34.9 and 36.9 mg/g) was recorded in the untreated control in both seasons. Respecting the interaction between the two studied factors (cultivars and fertilization), Misr-1 achieved the highest values of total chlorophyll content (48.8 and 49.4) during both seasons, with the application of 240 kg Mn-N/ha + 14 L Nan-N/ha.

#### 3.2.2. Plant Height

The results in Table 2 display moderate differences in plant height between the two cultivars in both seasons. Concerning the effects of fertilization treatments on plant height, significant differences were detected among the applied fertilization levels in both seasons (Table 3). The foliar application of nano-nitrogen fertilizer at 14 L/ha combined with mineral fertilizer at 240 kg/ha recorded the tallest plants (86.2 and 86.5 cm), while the untreated control recorded the shortest ones (77.3 and 77.3 cm) during both seasons. Otherwise, the two evaluated cultivars displayed no statistically significant differences in plant height. Regarding the interactive effect of wheat cultivars and fertilization treatments, the presented data in Table 2 revealed that no significant interaction was found in plant height during the two growing seasons.

#### 3.2.3. Number of Tillers per Square Meter

The results in Table 3 demonstrated minor differences between wheat cultivars in the number of tillers/m^2^ during the first season, while significant differences were observed in the second season. The Misr-1 cultivar produced the highest number of tillers/m^2^ (409.37 and 320.61) compared to Gemaiza-11 cv (396.8 and 301.31) in the consecutive seasons, respectively. The number of tillers/m^2^ was significantly altered owing to the impact of applied fertilization treatments for each of the two seasons (Table 3). Wheat cultivars fertilized with a rate of 240 kg Mn-N/ha + 14 L Nan-N/ha produced the highest number of tillers/m^2^ (424.82 and 354.32 tillers/m^2^) compared to untreated control (375.12 and 260.48 tillers/m^2^) during both seasons, respectively. The number of tillers/m^2^ increased in dependence on the combination of mineral N and nano-fertilizers—the interaction between the studied variables for the number of tillers/m^2^, Table 3. Misr-1 recorded the highest values for the number of tillers/m^2^ (430.73 and 366.76 tillers/m^2^) under the combined treatment of 240 kg Mn-N/ha + 14 L Nan-N/ha during the two seasons, respectively.

#### 3.2.4. Spike Length (cm)

The results in Table 4 show that the Misr-1 cultivar recorded a longer spike length (11.09 and 12.01 cm) compared to Gemaiza-11 (10.37 and 10.63 cm) during both seasons. The spike length was significantly affected by applying different levels of nitrogen fertilization. Among nitrogen treatments, the combined treatment of 120 kg Mn-N/ha + 14 L Nan-N/ha contributed the longest spikes (11.50 and 13.50 cm) compared to the untreated control (10.02 and 8.40 cm) in the two seasons, respectively. The interaction between wheat cultivars and fertilization treatments was significant for spike length (Table 5). Misr-1 under the application of Mn-N 120 kg/ha + 14 L Nan-N/ha produced the longest spikes compared to the other interactions, recording 11.80 and 15.43 cm during the two seasons, respectively.

#### 3.2.5. Number of Spikelets/Spike

The number of spikelets per spike is an important attribute of grain yield. In the first season, no significant difference was observed between the two cultivars, whereas it was statistically significant in the second season (Table 5). In the first and second seasons, respectively, Misr-1 produced the highest number of spikelets/spike (18.51 and 17.81) compared to Gemaiza-11 (18.48 and 15.85). The results presented in (Table 5) show that the fertilization treatments significantly impacted the number of spikelets per spike in both seasons. The application of 120 kg Mn-N/ha in combination with 14 L Na-N/ha provided the highest number of spikelets/spike (19.14 and 19.35), while the untreated control recorded the lowest ones (17.14 and 15.57). The interaction effect between the cultivars and the fertilization treatments showed that Misr-1 had the maximum number of spikelets/spike (19.20 and 19.93) in the first and second seasons, respectively, under 120 Mn-N kg/ha + 14 L Nan-N/ha compared to the remaining interactions. 

#### 3.2.6. Spike Weight

The wheat cultivars did not significantly affect the spike weight during the second season (Table 6). In contrast, spike weight was affected by applying different levels of mineral or nano-nitrogen fertilizers. The application of 120 kg Mn-N/ha combined with 14 L Nan-N/ha provided the heaviest weight of spike (12.78 and 12.00 g) compared to untreated control (10.23 and 10.00 g) during the two seasons, respectively. The interaction between wheat cultivars and fertilization treatments was insignificant for spike weight during the first season, while it was significant during the second one (Table 6). The maximum spike weight (13.56 and 12.33 g) was produced by the Misr-1 cultivar fertilized with 120 kg Mn-N/ha + 14 L Nan-N/ha in the first and second seasons, respectively. 

#### 3.2.7. 1000-Grain Weight

The data in Table 7 illustrated that the differences between the two studied cultivars were significant in terms of 1000-grain weight in the two seasons. Misr-1 was superior to Gemaiza-11 in 1000-grain weight during both seasons. The average weight was 54.0 and 54.3 g for Misr-1 and 49.6 and 53.4 g for Gemaiza-11 in the two seasons, respectively. The fertilization treatments significantly influenced the 1000-grain weight in the two seasons. Among the treatments, the heaviest 1000-grain weight (55.8 and 57.4 g) was recorded with the treatment of 120 kg Mn-N with 14 L Nan-N/ha compared with the untreated control (47.6 and 45.5 g) during the two seasons, respectively. The interaction between Misr-1 and fertilization application of 120 kg Mn-N/ha along with 14 L Nan-N/ha provided the heaviest grain weight (58.1 and 57.6 g) during the first and second seasons, respectively. 

#### 3.2.8. Grain Yield

Cultivars significantly affected grain yield during the two seasons (Table 8). Misr-1 achieved a higher grain yield than Gemaiza-11 in both seasons. Moreover, the two seasons significantly affected grain yield by applying different levels of nano- and mineral nitrogen fertilizers (Table 8). Among the applied fertilization treatments, 120 kg Mn-N/ha with 14 L Nan-N/ha produced the maximum grain yield compared with the untreated control in the two seasons. The interaction between the two studied factors, cultivars and fertilization applications, was statistically significant in both seasons. Misr-1 treated with a combination of 120 kg Mn-N/ha + 14 L Nan-N/ha achieved the highest values for grain yield in the two studied seasons.

#### 3.2.9. Straw Yield

The data in Appendix A declared that the two evaluated cultivars exhibited a significant difference in straw yield/ha in both seasons, with the superiority of Misr-1 over Gemaiza-11. Moreover, the straw yield was significantly increased by increasing nitrogen levels. The maximum straw yield was recorded by applying 240 kg Mn-N/ha + 14 L Nan-N/ha, while the untreated control obtained the minimum yield in both seasons. The interaction between cultivars and fertilization applications was not significant for straw yield during the first season, while it was significant for this trait during the second one (Table 9). The interaction between the Misr-1 and 240 kg Mn-N/ha + 7 L Nan-N/ha recorded the highest value for straw yield during the second season. 

#### 3.2.10. Biological Yield

The total biomass of the crop was not significantly influenced by cultivar performance during either season (Table 10). The biomass content was 22,223 and 21,447 kg/ha for Gemaiza-11 and Misr-1 in the first season, respectively, but decreased in the second. Otherwise, the applied nitrogen sources exhibited significant differences in the total biomass of wheat during the two seasons (Table 10). The combined treatment of 240 kg Mn-N/ha + 14 L Nan-N/ha produced the highest wheat biomass compared to the other applications in the first season. The application of 240 kg Mn-N/ha + 7 L Nan-N/ha exhibited the highest biological yield in the second season.

Regarding the interaction between wheat cultivars and fertilization treatments, there was no significant effect on biological yield in the first season. However, the second season experienced a significant effect (Table 10). The combination of 120kg Mn-N/ha + 14 L/ha Nan-N displayed the highest biological yield in both cultivars, with no statistically significant difference between 240 kg Mn-N/ha + 7L Nan-N/ha and 240 kg Mn-N/ha + 14L Nan-N/ha.

### 3.3. Nitrogen, Potassium, and Sodium Concentrations

Data in Table 9, Table 11 and Table 12 revealed that mineral contents of nitrogen, potassium, and sodium were significantly affected by the cultivar in the first and second seasons. The Misr-1 cultivar had higher nitrogen (1.30 and 1.91 mg/L) and potassium (9.87 and 9.81 mg/L) than Gemaiza-11 in the first and second seasons, respectively. On the other hand, Gemaiza-11 recorded a higher sodium content (6.07 and 5.97 mg/L) than Misr-1 (5.74 and 5.79 mg/L) throughout the first and second seasons, respectively. 

Regarding nitrogen fertilization treatments, the highest nitrogen (1.82 and 2.27 mg/L) and potassium (10.53 and 10.83 mg/L) levels were obtained by the combined treatment of 120 kg Mn-N/ha with 14 L/ha for the two seasons, respectively. Otherwise, the untreated control had a high sodium concentration (6.28 and 6.78 mg/L) during the two seasons, respectively. The interaction between the two studied factors displayed significant differences in mineral content. The highest nitrogen (2.00 and 2.60 mg/L) and potassium (10.93 and 10.87 mg/L) contents were recorded by Misr-1 under treatment with 120 kg Mn-N/ha + 14 L Nan-N/ha during the two seasons, respectively. Moreover, the highest concentrations of sodium (6.41 and 7.27 mg/L) were recorded by untreated Gemaiza-11 during the two seasons. 

## 4. Discussion 

Nitrogen (N) is an essential mineral nutrient required in high concentrations for wheat growth, productivity, and quality [43,44]. Accordingly, among the agricultural inputs, N fertilizer is a key concern. There is a change in global N supply and demand dynamics because of increasing consumption and high prices. The high cost of chemical nitrogenous fertilizers and the low purchasing power of most farmers restrict the use of these fertilizers in proper amounts and hamper crop production. Besides, a substantial amount of the urea-N is lost through different mechanisms, including ammonia volatilization, denitrification, and leaching losses, causing environmental pollution problems, contamination of groundwater resources, soils, and water resources, and the over-accumulation of nitrogen in plant tissues, in addition to increasing production costs.

On the other hand, using nitrogen fertilization in nano form, especially chitosan nanoparticles, determines the exact amounts of nitrogen required for plants in an available way, is eco-friendly, slow-releases pollution, and increases microbial activity according to [16] and Singh et al., [19]. Applying nano-nitrogen fertilizers with reduced doses of mineral N fertilizers can boost the productivity of cereal crops by achieving the maximum amount of available nitrogen [20]. 

In this present study, the combined application of 120 kg Mn-N/ha with 14 L Nan-N/ha recorded the maximum total chlorophyll content, the number of tillers/m^2^, spike length, number of spikelets per spike, spike weight, 1000-grain weight, and natural straw and grain yields (kg/ha) under the two growing seasons. This increase in grain yield and its components may be due to the nano-nitrogen fertilizers’ positive role in increasing nitrogen use efficiency (NUE) and reducing its application doses. These results are in agreement with those obtained by Seleem and Abd El-Dayem [45]; Mosaad and Fouda [46]; Bavar et al. [47]; Dimkpa and Bindraban [48]; Morsy et al. [49]; Juarez-Maldonado et al. [50]; Ayman et al. [51]; Qureshi et al. [52] and Yaseen et al. [53]. Otherwise, Chandini et al. [54] revealed that the application of mineral fertilizers like N in excess amounts has significant effects on both the soil and the groundwater due to the leaching of the remaining minerals into the soil and their unfavorable contribution to the air, thus having negative effects on both the sustainability and productivity of crops. Many applications have been previously reported for chitosan nanoparticles due to a positive surface charge increasing affinity towards biological membranes, resulting in enhanced reactivity with biological systems and controlled-release agents for nitrogen, phosphorus, and potassium fertilizers [55,56]. 

Chitosan NPs (90 ppm) can mitigate adverse effects of drought in wheat under drought stress [57]. In this context, Abdel-Aziz et al. [58] stated that Chitosan-NPK nanoparticles were easily applied to wheat leaf surfaces and entered the stomata via gas uptake, avoiding direct interaction with soil systems. Their results revealed that nano fertilizer was taken up and transported through phloem tissues. Treating wheat plants grown in sandy soil with nano chitosan-NPK fertilizer induced significant increases in harvest index, crop index, and mobilization index of the determined wheat yield variables when comparing positive and negative controls. The life cycle of the nano-fertilized wheat plants was shorter than that of normally fertilized wheat plants. Consequently, accelerating plant growth and productivity by the application of nano-fertilizer can open new perspectives in agricultural practice. Additionally, Li et al. [59] found that chitosan NPs induced auxin-related gene expression, accelerated indole-3-acetic acid (IAA) biosynthesis and transport, and reduced IAA oxidase activity, resulting in an increase in IAA concentration in wheat shoots and roots. These results suggest that chitosan NPs have a positive effect on wheat germination and seedling growth at a lower concentration than CS due to higher adsorption on the surface of wheat seeds. In contrast, the limitations of using chitosan NPs at the current level of knowledge does not allow a fair assessment of the pros and cons arising from using chitosan-based nano-pesticides in agriculture. It is necessary to better understand the fate and effects of such products after their application. The suitability of current regulations should also be analyzed to implement refinements if needed. Another major hurdle in sustainable agriculture is the removal of harmful contaminants from the soil. The unique properties of chitosan nanoparticles may prove to be useful in environmental detection, sensing, and remediation systems. Besides, the response of plants to nano-fertilizer varies with the type of plant species, their growth stages, and the nature of nanomaterials [60].

Cultivars’ yield potential and ability to deliver N fertilizers differ due to genetic differences and their adaptability to environmental conditions which are reflected in growth attributes and yield traits. The cultivars’ performance in an environment produces genotypic and environmental effects and effects from the interaction between the genotype and environmental effects [61]. 

The current study showed significant differences between the two cultivars, Misr-1 and Gemaiza-11, in almost all studied yield attributes. These effects give the cultivars their phenotypic values, which are used to select the higher-yielding and more stable cultivars in different environments. In the present study, the interaction between the Misr-1 cultivar and the application of 120 kg Mn-N/ha with 14 L Nan-N/ha exhibited the highest values of most of the studied growth, yield, and quality parameters, including the concentration of nitrogen and potassium. In this respect, Juarez-Maldonado et al. [50]; Ayman et al., [51]; Qureshi et al. [52]; and Yaseen et al. [53] elucidated a significant interaction between cultivars and their genotypes and fertilization applications. Several studies obtained significant differences among cultivars for total chlorophyll, number of tillers/m^2^, spike length, number of spikelets per spike [62,63,64], plant height [36,65], spike length [66,67,68], 1000-grain weight, and grain [69].

## 5. Conclusions

Chitosan NPs fabricated by *Penicillium oxalicum* are a cheap and valuable source of nitrogen. Increasing the availability of nitrogen is an important approach to increasing nutrient efficiency, boosting plant nutrition, improving yield traits, and minimizing soil contamination. The application of chitosan NPs combined with mineral nitrogen fertilization enhanced the productivity of wheat plants. The foliar application of 120 kg Mn-N/ha + 14 L Nan-N/ha recorded the highest values for grain yield and most of its components as well as nitrogen and potassium concentrations. The evaluated cultivars displayed significant differences in most growth, yield, and quality parameters. Misr-1 exhibited the highest total chlorophyll content, spike length, 100-grain weight, and grain yield kg/ha as well as nitrogen and potassium concentrations. 

## Figures and Tables

**Figure 1 molecules-27-05640-f001:**
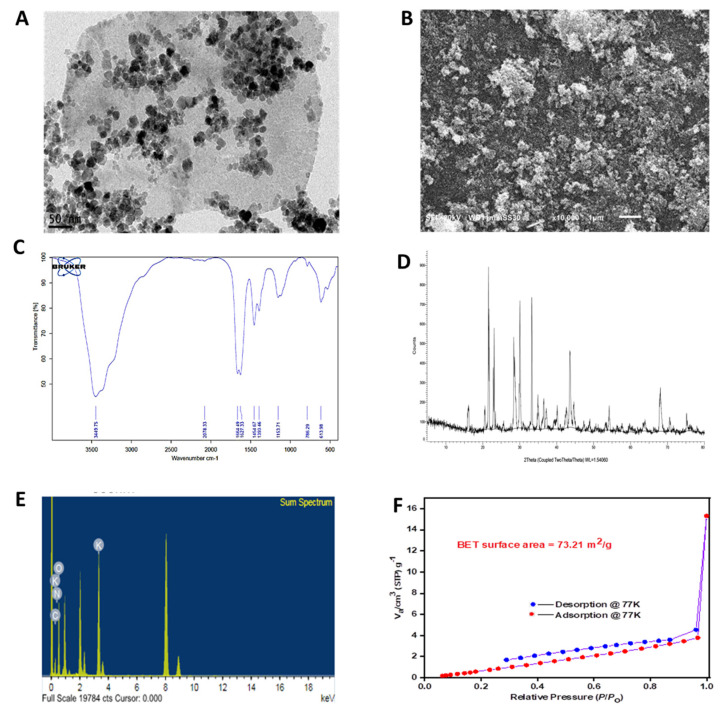
Characterization of chitosan nanoparticles fabricated by *Penicillium oxalicum* from shrimp waste (**A**) and (**B**) TEM and SEM images, (**C**) FTIR images for active groups in chitosan nanoparticles, (**D**) XRD of chitosan nanoparticles, © the energy dispersive X-ray (EDX) for chitosan nanoparticles (**E**) N_2_ adsorption-desorption isotherms of the fabricated chitosan nanoparticles (**F**).

**Table 1 molecules-27-05640-t001:** Impact of nano and mineral nitrogen fertilizers on total chlorophyll content of two wheat cultivars during 2019/2020 and 2020/2021 seasons.

Fertilization Source (F)		First Season		Mean (F)	Second Season		Mean (F)
		Cultivar (C)			Cultivar (C)		
		Gemaiza-11	Misr-1		Gemaiza-11	Misr-1	
Untreated-control (without N)		35.3 ± 0.95	34.5 ± 0.46	34.9 ± 0.63 ^F^	36.5 ± 0.46	37.3 ± 0.98	36.9 ± 0.54 ^E^
Mn-N (120 kg/ha)		44.0 ± 0.58	45.0 ± 0.69	44.5 ± 0.46 ^D^	43.0 ± 0.95	45.0 ± 0.58	44.0 ± 0.73 ^CD^
Mn-N (240 kg/ha)		45.3 ± 0.52	47.5 ± 0.61	46.4 ± 0.59 ^C^	44.5 ± 0.87	46.3 ± 0.92	45.4 ± 0.69 ^C^
Nan-N (7 L/ha)		38.5 ± 0.92	37.7 ± 0.49	38.1 ± 0.47 ^F^	39.9 ± 0.41	38.7 ± 0.95	39.3 ± 0.63 ^E^
Nan-N (14 L/ha)		43.6 ± 0.98	42.7 ± 0.52	43.2 ± 0.60 ^E^	41.8 ± 0.35	43.7 ± 0.69	42.7 ± 0.55 ^D^
Mn-N 120 kg/ha + Nan-N 7 L/ha		44.6 ± 0.87	46.0 ± 0.75	45.3 ±0.63 ^D^	44.5 ± 0.64	45.6 ± 0.46	45.1 ± 0.43 ^C^
Mn-N 120 kg/ha + Nan-N 14 L/ha		45.7 ± 0.64	47.8 ± 0.46	46.7 ± 0.59 ^C^	45.8 ± 0.67	45.9 ± 0.52	45.8 ± 0.39 ^C^
Mn-N 240 kg/ha + Nan-N 7 L/ha		48.0 ± 0.46	47.9 ± 0.52	47.9 ± 0.39 ^B^	46.2 ± 0.81	47.3 ± 0.49	46.7 ± 0.50 ^B^
Mn-N 240 kg/ha + Nan-N 14 L/ha		48.5 ± 0.59	48.8 ± 0.81	48.6 ± 0.46 ^A^	48.9 ± 0.37	49.4 ± 0.46	49.2 ± 0.41 ^A^
Mean (C)		43.8 ± 0.82 ^B^	44.2 ± 0.94 ^A^		43.5 ± 0.71 ^B^	44.3 ± 0.76 ^A^	
ANOVA	df						
Cultivar (C)	1	0.007			0.034		
Fertilization Source (F)	8	<0.001			0.001		
F × C	8	0.307			0.314		

Different uppercase letters indicate significant difference among the evaluated cultivars or fertilization source at *p* < 0.05.

**Table 2 molecules-27-05640-t002:** Impact of nano and mineral nitrogen fertilizers on the plant height (cm) of two wheat cultivars during the 2019/2020 and 2020/2021 seasons.

Fertilization source (F)		First Season		Mean (F)	Second Season		Mean (F)
		Cultivar (C)			Cultivar (C)		
		Gemaiza-11	Misr-1		Gemaiza-11	Misr-1	
Untreated-control (without N)		77.2 ± 2.8	77.4 ± 2.1	77.3 ± 1.9 ^F^	79.3 ± 3.1	75.2 ± 2.5	77.3 ± 2.0 ^C^
Mn-N (120 kg/ha)		82.7 ± 2.6	81.9 ± 2.2	82.3 ± 2.5 ^D^	84.2 ± 1.7	80.0 ± 2.9	82.1 ± 2.5 ^B^
Mn-N (240 kg/ha)		83.0 ± 3.0	83.4 ± 3.2	83.2 ± 2.8 ^C^	86.8 ± 3.0	80.1 ± 1.6	83.4 ± 2.4 ^AB^
Nan-N (7 L/ha)		83.3 ± 2.8	79.5 ± 2.1	81.4 ± 1.6 ^E^	76.0 ± 2.5	79.8 ± 1.8	77.9 ± 1.8 ^C^
Nan-N (14 L/ha)		82.9 ± 2.5	81.5 ± 1.9	82.2 ± 1.2 ^D^	82.1 ± 3.1	77.4 ± 1.8	79.8 ± 2.2 ^BC^
Mn-N 120 kg/ha + Nan-N 7 L/ha		84.4 ± 2.6	82.3 ± 1.8	83.4 ± 1.7 ^C^	82.4 ± 2.5	78.7 ± 1.2	80.6 ± 1.8 ^BC^
Mn-N 120 kg/ha + Nan-N 14 L/ha		85.0 ± 1.9	81.7 ± 3.2	83.5 ± 2.6 ^C^	79.3 ± 1.7	81.9 ± 1.3	80.6 ± 1.7 ^BC^
Mn-N 240 kg/ha + Nan-N 7 L/ha		86.1 ± 2.6	81.9 ± 1.4	84.0 ± 1.3 ^B^	87.3 ± 1.8	83.5 ± 2.6	85.4 ± 1.6 ^AB^
Mn-N 240 kg/ha + Nan-N 14 L/ha		87.6 ± 2.5	84.8 ± 2.7	86.2 ± 1.7 ^A^	88.5 ± 1.8	84.4 ± 1.4	86.5 ± 1.9 ^A^
Mean (C)		83.6 ± 1.5	81.6 ± 1.7		82.9 ± 1.2	80.1 ± 1.5	
ANOVA	df						
Cultivar (C)	1	0.917			0.138		
Fertilization Source (F)	8	0.038			0.005		
F × C	8	0.577			0.571		

Different uppercase letters indicate significant differences among the evaluated fertilization sources at *p* < 0.05.

**Table 3 molecules-27-05640-t003:** Impact of nano- and mineral nitrogen fertilizers on number of tillers per square meter of two wheat cultivars during the 2019/2020 and 2020/2021 seasons.

Fertilization Source (F)		First Season		Mean (F)	Second Season		Mean (F)
		Cultivar (C)			Cultivar (C)		
		Gemaiza-11	Misr-1		Gemaiza-11	Misr-1	
Untreated-control (without N)		356.0 ± 1.7 ^e^	394.23 ± 2.0 ^c^	375.12 ± 2.8 ^C^	238.52 ± 2.8 ^g^	282.44 ± 3.1 ^ef^	260.48 ± 2.7 ^G^
Mn-N (120 kg/ha)		400.10 ± 2.5 ^bcd^	408.20 ± 3.1 ^b^	404.15 ± 3.1 ^B^	301.35 ± 1.8 ^de^	289.57 ± 1.2 ^ef^	295.46 ± 1.4 ^EF^
Mn-N (240 kg/ha)		402.54 ± 3.1 ^bcd^	412.30 ± 2.8 ^b^	407.42 ± 3.2 ^AB^	276.28 ± 2.5 ^f^	361.85 ± 2.8 ^ab^	319.07 ± 2.2 ^CD^
Nan-N (7 L/ha)		406.36 ± 2.5 ^abcd^	361.78 ± 1.6 ^d^	384.07 ± 2.4 ^BC^	286.75 ± 2.6 ^ef^	282.26 ± 2.7 ^ef^	284.51 ± 3.1 ^F^
Nan-N (14 L/ha)		393.23 ± 3.0 ^d^	397.22 ± 1.4 ^c^	395.23 ± 3.1 ^B^	305.63 ± 2.6 ^de^	282.75 ± 2.8 ^ef^	294.19 ± 2.6 ^EF^
Mn-N 120 kg/ha + Nan-N 7 L/ha		396.25 ± 1.7 ^bcd^	420.10 ± 3.1 ^ab^	408.18 ± 1.7 ^AB^	268.10 ± 2.7 ^f^	349.13 ± 2.8 ^ab^	308.62 ± 2.8 ^DE^
Mn-N 120 kg/ha + Nan-N 14 L/ha		390.81 ± 2.6 ^d^	429.32 ± 1.7 ^a^	410.07 ± 2.6 ^AB^	337.57 ± 3.0 ^bc^	321.43 ± 3.1 ^cd^	329.50 ± 3.1 ^BC^
Mn-N 240 kg/ha + Nan-N 7 L/ha		406.64 ± 1.9 ^abcd^	430.47 ± 2.4 ^a^	418.56 ± 2.0 ^A^	346.72 ± 3.3 ^ab^	349.32 ± 1.8 ^ab^	348.02 ± 1.8 ^AB^
Mn-N 240 kg/ha + Nan-N 14 L/ha		418.90 ± 2.3 ^abc^	430.73 ± 3.2 ^a^	424.82 ± 2.1 ^A^	341.88 ± 2.6 ^ab^	366.76 ± 2.1 ^a^	354.32 ± 1.9 ^A^
Mean (C)		396.8 ± 1.5 ^B^	409.37 ± 1.6 ^A^		300.31 ± 2.1 ^B^	320.61 ± 1.8 ^A^	
ANOVA	df						
Cultivar (C)	1	0. 046			0.024		
Fertilization Source (F)	8	0.005			0.002		
F × C	8	0.047			0.001		

Different uppercase letters indicate significant differences among the evaluated cultivars or fertilization sources at *p* < 0.05, while different lowercase letters indicate significant differences among their interactions.

**Table 4 molecules-27-05640-t004:** Impact of nano- and mineral nitrogen fertilizers on the spike length (cm) of two wheat cultivars during the 2019/2020 and 2020/2021 seasons.

Fertilization Source (F)		First Season		Mean (F)	Second Season		Mean (F)
		Cultivar (C)			Cultivar (C)		
		Gemaiza-11	Misr-1		Gemaiza-11	Misr-1	
Untreated-control (without N)		9.37 ± 0.23 ^g^	10.67 ± 0.67 ^cde^	10.02 ± 0.39 ^D^	8.00 ± 0.43 ^i^	8.80 ± 0.52 ^hi^	8.40 ± 0.58 ^F^
Mn-N (120 kg/ha)		10.20 ± 0.45 ^e^	11.27 ± 0.44 ^ab^	10.73 ± 0.52 ^B^	12.00 ± 0.62 ^cde^	11.53 ± 0.67 ^def^	11.77 ± 0.43 ^C^
Mn-N (240 kg/ha)		10.40 ± 0.70 ^df^	11.40 ± 0.32 ^ab^	10.90 ± 0.48 ^B^	11.20 ± 0.57 ^efg^	13.00 ± 0.52 ^c^	12.10 ± 0.91 ^BC^
Nan-N (7 L/ha)		10.60 ± 0.63 ^de^	9.87 ± 0.62 ^fg^	10.23 ± 0.36 ^C^	10.53 ± 0.48 ^fg^	10.00 ± 0.58 ^gh^	10.27 ± 0.58 ^D^
Nan-N (14 L/ha)		10.40 ± 0.84 ^def^	11.27 ± 0.51 ^ab^	10.83 ± 0.52 ^B^	10.77 ± 0.63 ^fg^	12.67 ± 0.49 ^cd^	11.72 ± 0.75 ^C^
Mn-N 120 kg/ha + Nan-N 7 L/ha		10.53 ± 0.52 ^de^	10.87 ± 0.38 ^bcd^	10.70 ± 0.72 ^B^	10.10 ± 0.58 ^g^	10.53 ± 0.49 ^f^	10.32 ± 0.42 ^D^
Mn-N 120 kg/ha + Nan-N 14 L/ha		11.20 ± 0.72 ^bc^	11.80 ± 0.47 ^a^	11.50 ± 0.42 ^A^	11.57 ± 0.77 ^def^	15.43 ± 0.79 ^a^	13.50 ± 0.52 ^A^
Mn-N 240 kg/ha + Nan-N 7 L/ha		10.47 ± 0.88 ^de^	11.33 ± 0.60 ^ab^	10.90 ± 0.60 ^B^	10.87 ± 0.92 ^efg^	14.90 ± 0.87 ^b^	12.88 ± 0.89 ^AB^
Mn-N 240 kg/ha + Nan-N 14 L/ha		10.20 ± 0.62 ^ef^	11.33 ± 0.58 ^ab^	10.77 ± 0.72 ^B^	10.67 ± 0.72 ^fg^	11.23 ± 0.80 ^efg^	10.95 ± 0.73 ^CD^
Mean (C)		10.37 ± 0.75 ^B^	11.09 ± 0.61 ^A^		10.63 ± 0.54 ^B^	12.01 ± 0.48 ^A^	
ANOVA	df						
Cultivar (C)	1	0.017			0.047		
Fertilization Source (F)	8	0.002			<0.001		
F × C	8	0.013			0.003		

Different uppercase letters indicate significant differences among the evaluated cultivars or fertilization source at *p* < 0.05, while different lowercase letters indicate significant differences among their interactions.

**Table 5 molecules-27-05640-t005:** Impact of nano- and mineral nitrogen fertilizers on number of spikelets/spike of wheat cultivars during the 2019/2020 and 2020/2021 seasons.

Fertilization Source (F)		First Season		Mean (F)	Second Season		Mean (F)
		Cultivar (C)			Cultivar (C)		
		Gemaiza-11	Misr-1		Gemaiza-11	Misr-1	
Untreated-control (without N)		16.80 ± 0.57 ^de^	17.47 ± 0.49 ^cd^	17.14 ± 0.82 ^C^	14.87 ± 0.92 ^h^	16.27 ± 0.74 ^fg^	15.57 ± 0.84 ^F^
Mn-N (120 kg/ha)		18.73 ± 0.79 ^b^	18.53 ± 0.52 ^bc^	18.63 ± 0.48 ^AB^	16.73 ± 0.80 ^f^	16.07 ± 0.92 ^fg^	16.40 ± 0.56 ^DEF^
Mn-N (240 kg/ha)		18.87 ± 0.67 ^b^	18.730.62± ^b^	18.80 ± 0.55 ^AB^	18.20 ± 0.59 ^cd^	18.40 ± 0.68 ^bcd^	18.30 ± 0.67 ^B^
Nan-N (7 L/ha)		19.20 ± 0.48 ^b^	16.27 ± 0.69 ^e^	17.74 ± 0.60 ^BC^	16.87 ± 0.48 ^ef^	15.00 ± 0.70 ^h^	15.94 ± 0.74 ^EF^
Nan-N (14 L/ha)		16.73 ± 0.62 ^de^	20.53 ± 0.51 ^a^	18.63 ± 0.72 ^AB^	15.27 ± 0.83 ^g^	18.07 ± 0.68 ^cde^	16.67 ± 0.52 ^CDE^
Mn-N 120 kg/ha + Nan-N 7 L/ha		18.93 ± 0.90 ^b^	18.47 ± 0.46 ^bc^	18.70 ± 0.49 ^AB^	16.33 ± 0.57 ^fg^	18.70 ± 0.59 ^abc^	17.52 ± 0.60 ^BC^
Mn-N 120 kg/ha + Nan-N 14 L/ha		19.07 ± 0.82 ^b^	19.20 ± 0.54 ^b^	19.14 ± 0.54 ^A^	18.77 ± 0.81 ^abc^	19.93 ± 0.60 ^a^	19.35 ± 0.49 ^A^
Mn-N 240 kg/ha + Nan-N 7 L/ha		18.93 ± 0.73 ^b^	19.20 ± 0.60 ^b^	19.07 ± 0.63 ^A^	18.33 ± 0.64 ^bcd^	19.47 ± 0.90 ^ab^	18.90 ± 0.91 ^A^
Mn-N 240 kg/ha + Nan-N 14 L/ha		18.80 ± 0.78 ^b^	18.47 ± 0.87 ^bc^	18.64 ± 0.51 ^AB^	17.27 ± 0.71 ^def^	18.40 ± 0.83 ^bcd^	17.84 ± 0.73 ^B^
Mean (C)		18.48 ± 0.57	18.51 ± 0.64		15.85 ± 0.63 ^B^	17.81 ± 0.56 ^A^	
ANOVA	df						
Cultivar (C)	1	0.938			0.016		
Fertilization Source (F)	8	0.036			< 0.001		
F × C	8	0.001			< 0.001		

Different uppercase letters indicate significant differences among evaluated cultivars or fertilization sources at *p* < 0.05, while different lowercase letters indicate significant differences among their interactions.

**Table 6 molecules-27-05640-t006:** Impact of nano- and mineral nitrogen fertilizers on the spike weight (g) of two wheat cultivars during the 2019/2020 and 2020/2021 seasons.

Fertilization Source (F)		First Season		Mean (F)	Second Season		Mean (F)
		Cultivar (C)			Cultivar (C)		
		Gemaiza-11	Misr-1		Gemaiza-11	Misr-1	
Untreat-control (without N)		10.78 ± 0.73	9.67 ± 0.67	10.23 ± 48 ^C^	10.00 ± 0.87 ^f^	10.00 ± 0.52 ^f^	10.00 ± 0.75 ^E^
Mn-N (120 kg/ha)		11.22 ± 0.85	12.56 ± 0.42	11.89 ± 0.52 ^B^	10.05 ± 0.62 ^f^	10.07 ± 0.74 ^f^	10.06 ± 0.87 ^DE^
Mn-N (240 kg/ha)		12.33 ± 1.00	12.44 ± 0.71	12.39 ± 0.58 ^A^	10.10 ± 0.49 ^f^	11.20 ± 0.84 ^c^	10.65 ± 0.92 ^B^
Nan-N (7 L/ha)		9.67 ± 0.94	12.56 ± 0.81	11.12 ± 0.92 ^B^	10.30 ± 0.84 ^ef^	10.30 ± 0.54 ^ef^	10.30 ± 0.67 ^CD^
Nan-N (14 L/ha)		11.78 ± 0.85	11.67 ± 0.57	11.73 ± 0.54 ^B^	10.30 ± 0.75 ^ef^	10.70 ± 0.68 ^d^	10.50 ± 0.76 ^BC^
Mn-N 120 kg/ha + Nan-N 7 L/ha		11.44 ± 0.52	12.78 ± 0.84	12.11 ± 0.95 ^AB^	10.50 ± 0.57 ^de^	10.09 ± 0.64 ^f^	10.30 ± 0.92 ^CD^
Mn-N 120 kg/ha + Nan-N 14 L/ha		12.00 ± 0.92	13.56 ± 0.72	12.78 ± 0.72 ^A^	11.67 ± 0.80 ^b^	12.33 ± 0.39 ^a^	12.00 ± 1.02 ^A^
Mn-N 240 kg/ha + Nan-N 7 L/ha		11.67 ± 0.54	13.44 ± 0.86	12.56 ± 0.84 ^A^	10.00 ± 0.92 ^f^	11.33 ± 0.81 ^bc^	10.67 ± 0.92 ^B^
Mn-N 240 kg/ha + Nan-N 14 L/ha		11.33 ± 0.84	13.33 ± 0.81	12.33 ± 0.66 ^A^	10.03 ± 0.64 ^f^	10.04 ± 0.69 ^f^	10.04 ± 0.82 ^DE^
Mean (C)		11.36 ± 0.75 ^B^	12.44 ± 0.87 ^A^		10.19 ± 0.57	10.52 ± 0.91	
ANOVA	df						
Cultivar (C)	1	0.049			0.121		
Fertilization Source (F)	8	0.046			<0.001		
F × C	8	0.294			0.003		

Different uppercase letters indicate significant differences among the evaluated cultivars or fertilization sources at *p* < 0.05, while different lowercase letters indicate significant differences among their interactions.

**Table 7 molecules-27-05640-t007:** Impact of nano- and mineral nitrogen fertilizers on the 1000-grain weight (g) of two wheat cultivars during the 2019/2020 and 2020/2021 seasons.

Fertilization Source (F)		First Season		Mean (F)	Second Season		Mean (F)
		Cultivar (C)			Cultivar (C)		
		Gemaiza-11	Misr-1		Gemaiza-11	Misr-1	
Untreated-control (without N)		49.7 ± 2.01 ^de^	45.4 ± 1.97 ^fg^	47.6 ± 1.42 ^E^	45.4 ± 1.28 ^f^	45.5 ± 1.54 ^f^	45.5 ± 1.84 ^I^
Mn-N (120 kg/ha)		52.0 ± 1.62 ^cd^	54.5 ± 1.84 ^b^	53.3 ± 1.97 ^B^	51.1 ± 1.87 ^e^	52.3 ± 2.32 ^de^	51.7 ± 1.65 ^G^
Mn-N (240 kg/ha)		51.5 ± 1.51 ^cd^	53.8 ± 2.01 ^bc^	52.7 ± 2.09 ^BC^	56.7 ± 1.97 ^ab^	56.7 ± 2.64 ^ab^	56.7 ± 2.31 ^D^
Nan-N (7 L/ha)		47.1 ± 2.17 ^f^	54.4 ± 2.08 ^b^	50.8 ± 1.24 ^D^	50.1 ± 2.1 ^e^	52.6 ± 1.95 ^cd^	51.4 ± 2.05 ^H^
Nan-N (14 L/ha)		47.7 ± 1.18 ^ef^	54.9 ± 1.21 ^b^	51.3 ± 1.98 ^D^	53.0 ± 1.87 ^cde^	55.5 ± 1.87 ^abc^	54.3 ± 1.84 ^F^
Mn-N 120 kg/ha + Nan-N 7 L/ha		44.5 ± 1.95 ^g^	57.4 ± 1.50 ^a^	51.0 ± 2.1 ^D^	54.2 ± 2.12 ^bcd^	56.3 ± 2.34 ^ab^	55.3 ± 1.64 ^E^
Mn-N 120 kg/ha + Nan-N 14 L/ha		53.5 ± 1.87 ^bc^	58.1 ± 1.63 ^a^	55.8 ± 1.5 ^A^	57.2 ± 1.98 ^a^	57.6 ± 2.31 ^a^	57.4 ± 2.34 ^B^
Mn-N 240 kg/ha + Nan-N 7 L/ha		50.6 ± 2.12 ^d^	54.7 ± 1.05 ^b^	52.7 ± 1.97 ^BC^	57.4 ± 1.83 ^a^	56.8 ± 1.65 ^ab^	57.1 ± 2.08 ^C^
Mn-N 240 kg/ha + Nan-N 14 L/ha		49.9 ± 2.08 ^de^	53.0 ± 1.79 ^bc^	51.5 ± 2.0 ^CD^	55.1 ± 2.51 ^abcd^	56.1 ± 1.75 ^ab^	55.6 ± 1.74 ^A^
Mean (C)		49.6 ± 1.51 ^B^	54.0 ± 1.73 ^A^		53.4 ± 1.92 ^B^	54.3 ± 1.67 ^A^	
ANOVA	df						
Cultivar (C)	1	0.034			0.046		
Fertilization Source (F)	8	0.003			<0.001		
F × C	8	0.002			0.048		

Different uppercase letters indicate significant differences among the evaluated cultivars or fertilization sources at *p* < 0.05, while different lowercase letters indicate significant differences among their interactions.

**Table 8 molecules-27-05640-t008:** Impact of nano- and mineral nitrogen fertilizers on the grain yield (kg/ha) of two wheat cultivars during the 2019/2020 and 2020/2021 seasons.

Fertilization Source (F)		First Season		Mean (F)	Second Season		Mean (F)
		Cultivar (C)			Cultivar (C)		
		Gemaiza-11	Misr-1		Gemaiza-11	Misr-1	
Untreated-control (without N)		4698 ± 232 ^h^	4673 ± 219 ^h^	4686 ± 147 ^F^	3898 ± 173 ^g^	4660 ± 193 ^g^	4279 ± 188 ^F^
Mn-N (120 kg/ha)		5702 ± 180 ^fg^	6248 ± 201 ^cde^	5975 ± 115 ^D^	6234 ± 294 ^cde^	5054 ± 192 ^fg^	5644 ± 226 ^DE^
Mn-N (240 kg/ha)		6362 ± 229 ^bcde^	6820 ± 140 ^ab^	6591 ± 138 ^AB^	6997 ± 309 ^abc^	7112 ± 228 ^ab^	7054 ± 269 ^AB^
Nan-N (7 L/ha)		5905 ± 274 ^efg^	5512 ± 269 ^g^	5708 ± 213 ^E^	5562 ± 136 ^ef^	5321 ± 302 ^efg^	5442 ± 271 ^E^
Nan-N (14 L/ha)		6184 ± 141 ^def^	6692 ± 210 ^ab^	6438 ± 302 ^BC^	5956 ± 187 ^de^	6540 ± 176 ^bcd^	6248 ± 142 ^CD^
Mn-N 120 kg/ha + Nan-N 7 L/ha		6312 ± 158 ^bcde^	6400 ± 276 ^bcd^	6356 ± 241 ^BC^	6375 ± 269 ^bcde^	6731 ± 248 ^abcd^	6553 ± 301 ^BC^
Mn-N 120 kg/ha + Nan-N 14 L/ha		6540 ± 301 ^bcd^	7225 ± 240 ^a^	6883 ± 209 ^A^	7607 ± 249 ^a^	7747 ± 291 ^a^	7677 ± 308 ^A^
Mn-N 240 kg/ha + Nan-N 7 L/ha		6413 ± 169 ^bcde^	6769 ± 251 ^abc^	6591 ± 297 ^A^	6998 ± 243 ^abc^	7138 ± 305 ^ab^	7068 ± 268 ^AB^
Mn-N 240 kg/ha + Nan-N 14 L/ha		5982 ± 139 ^efg^	6350 ± 294 ^bcde^	6166 ± 218 ^CD^	5918 ± 212 ^de^	6478 ± 287 ^bcd^	6198 ± 306 ^CD^
Mean (C)		6011 ± 143 ^B^	6299 ± 169 ^A^		6172 ± 228 ^B^	6309 ± 252 ^A^	
ANOVA	df						
Cultivar (C)	1	0.011			0.038		
Fertilization Source (F)	8	<0.001			<0.001		
F × C	8	0.005			0.047		

Different uppercase letters indicate significant differences among the evaluated cultivars or fertilization sources at *p* < 0.05, while different lowercase letters indicate significant differences among their interactions.

**Table 9 molecules-27-05640-t009:** Impact of nano- and mineral nitrogen fertilizers on the nitrogen concentrations (Conc; mg/L) of two wheat cultivars during the 2019/2020 and 2020/2021 seasons.

Fertilization Source (F)		First Season		Mean (F)	Second Season		Mean (F)
		Cultivar (C)			Cultivar (C)		
		Gemaiza-11	Misr-1		Gemaiza-11	Misr-1	
Untreated-control (without N)		0.92 ± 0.053 ^g^	1.00 ± 0.063 ^fg^	0.96 ± 0.037 ^E^	1.16 ± 0.068 ^e^	1.56 ± 0.084 ^d^	1.36 ± 0.045 ^D^
Mn-N (120 kg/ha)		1.28 ± 0.049 ^cd^	1.14 ± 0.057 ^def^	1.21 ± 0.031 ^D^	1.75 ± 0.062 ^bcd^	1.77 ± 0.067 ^bcd^	1.76 ± 0.034 ^BC^
Mn-N (240 kg/ha)		1.36 ± 0.069 ^c^	1.32 ± 0.063 ^cd^	1.34 ± 0.043 ^C^	1.90 ± 0.051 ^bc^	1.96 ± 0.059 ^b^	1.93 ± 0.057 ^B^
Nan-N (7 L/ha)		0.92 ± 0.047 ^g^	1.04 ± 0.078 ^fg^	0.98 ± 0.051 ^E^	1.57 ± 0.039 ^d^	1.56 ± 0.076 ^d^	1.57 ± 0.049 ^C^
Nan-N (14 L/ha)		1.27 ± 0.043 ^cde^	1.16 ± 0.049 ^def^	1.22 ± 0.038 ^D^	1.88 ± 0.061 ^bc^	1.91 ± 0.081 ^bc^	1.90 ± 0.065 ^BC^
Mn-N 120 kg/ha + Nan-N 7 L/ha		1.00 ± 0.046 ^fg^	1.08 ± 0.041 ^efg^	1.04 ± 0.041 ^DE^	1.59 ± 0.056 ^d^	1.70 ± 0.049 ^cd^	1.65 ± 0.040 ^C^
Mn-N 120 kg/ha + Nan-N 14 L/ha		1.63 ± 0.047 ^b^	2.00 ± 0.045 ^a^	1.82 ± 0.029 ^A^	1.93 ± 0.048 ^bc^	2.60 ± 0.055 ^a^	2.27 ± 0.045 ^A^
Mn-N 240 kg/ha + Nan-N 7 L/ha		1.41 ± 0.067 ^c^	1.84 ± 0.035 ^a^	1.63 ± 0.032 ^B^	1.90 ± 0.072 ^bc^	2.40 ± 0.032 ^a^	2.15 ± 0.033 ^A^
Mn-N 240 kg/ha + Nan-N 14 L/ha		1.14 ± 0.071 ^def^	1.14 ± 0.030 ^def^	1.14 ± 0.034 ^D^	1.73 ± 0.069 ^bcd^	1.70 ± 0.039 ^cd^	1.72 ± 0.062 ^C^
Mean (C)		1.21 ± 0.037 ^B^	1.30 ± 0.031 ^A^		1.71 ± 0.045 ^B^	1.91 ± 0.037 ^A^	
ANOVA	df						
Cultivar (C)	1	0.015			0.024		
Fertilization Source (F)	8	0.003			0.008		
F × C	8	0.004			0.001		

Different uppercase letters indicate significant differences among the evaluated cultivars or fertilization sources at *p* < 0.05, while different lowercase letters indicate significant differences among their interactions.

**Table 10 molecules-27-05640-t010:** Impact of nano- and mineral nitrogen fertilizers on the biological yield (kg/ha) of two wheat cultivars during the 2019/2020 and 2020/2021 seasons.

Fertilization Source (F)		First Season		Mean (F)	Second Season		Mean (F)
		Cultivar (C)			Cultivar (C)		
		Gemaiza-11	Misr-1		Gemaiza-11	Misr-1	
Untreated-control (without N)		14,285 ± 550	15,238 ± 495	14,762 ± 408 ^E^	15,874 ± 457 ^e^	17,460 ± 635 ^de^	16,667 ± 623 ^D^
Mn-N (120 kg/ha)		20,953 ± 487	20,001 ± 509	20,477 ± 422 ^D^	19,047 ± 617 ^cd^	19,841 ± 347 ^bc^	19,444 ± 456 ^C^
Mn-N (240 kg/ha)		21,906 ± 524	21,906 ± 513	21,906 ± 347 ^C^	19,841 ± 550 ^bc^	20,636 ± 415 ^bc^	20,239 ± 317 ^BC^
Nan-N (7 L/ha)		19,683 ± 317	19,048 ± 476	19,366 ± 308 ^D^	21,430 ± 521 ^bc^	20,636 ± 629 ^bc^	21,033 ± 481 ^BC^
Nan-N (14 L/ha)		20,001 ± 492	20,001 ± 438	20,001 ± 354 ^D^	22,223 ± 489 ^ab^	22,224 ± 614 ^ab^	22,223 ± 572 ^B^
Mn-N 120 kg/ha + Nan-N 7 L/ha		23,811 ± 612	21,855 ± 611	22,833 ± 607 ^C^	22,224 ± 317 ^ab^	20,637 ± 640 ^b^	21,430 ± 657 ^BC^
Mn-N 120 kg/ha + Nan-N 14 L/ha		25,005 ± 357	24,370 ± 584	24,687 ± 455 ^B^	25,500 ± 428 ^a^	24,604 ± 317 ^a^	25,052 ± 395 ^A^
Mn-N 240 kg/ha + Nan-N 7 L/ha		26,351 ± 598	24,814 ± 642	25,582 ± 624 ^B^	25,398 ± 642 ^a^	26,059 ± 384 ^a^	25,729 ± 408 ^A^
Mn-N 240 kg/ha + Nan-N 14 L/ha		28,014 ± 587	25,791 ± 546	26,903 ± 607 ^A^	24,503 ± 658 ^a^	24,739 ± 635 ^a^	24,621 ± 528 ^A^
Mean (C)		22,223 ± 604	21,447 ± 662		21,781 ± 458	21,869 ± 518	
ANOVA	df						
Cultivar (C)	1	0.078			0.084		
Fertilization Source (F)	8	<0.001			<0.001		
F × C	8	0.759			0.014		

Different uppercase letters indicate significant differences among the evaluated fertilization sources at *p* < 0.05, while different lowercase letters indicate significant differences among their interactions.

**Table 11 molecules-27-05640-t011:** Impact of nano- and mineral nitrogen fertilizers on the potassium concentrations (Conc; mg/L) of two wheat cultivars during the 2019/2020 and 2020/2021 seasons.

Fertilization Source (F)		First Season		Mean (F)	Second Season		Mean (F)
		Cultivar (C)			Cultivar (C)		
		Gemaiza-11	Misr-1		Gemaiza-11	Misr-1	
Untreated-control (without N)		8.77 ± 0.91 ^fg^	8.68 ± 0.42 ^g^	8.73 ± 0.85 ^E^	8.91 ± 0.99 ^g^	8.96 ± 0.68 ^fg^	8.94 ± 0.84 ^E^
Mn-N (120 kg/ha)		9.39 ± 0.45 ^def^	9.78 ± 0.87 ^bcde^	9.59B ± 0.90 ^CD^	9.40 ± 0.84 ^efg^	9.50 ± 0.90 ^defg^	9.45 ± 0.89 ^CD^
Mn-N (240 kg/ha)		9.84 ± 0.39 ^bcde^	10.32 ± 0.97^ab^	10.08 ± 0.95 ^AB^	9.96 ± 0.74 ^cde^	10.52 ± 0.82 ^abc^	10.2 4 ± 0.67 ^B^
Nan-N (7 L/ha)		8.96 ± 0.48 ^fg^	9.15 ± 0.78 ^efg^	9.06 ± 0.82 ^DE^	8.96 ± 0.85 ^fg^	8.99 ± 0.92 ^fg^	8.98 ± 0.84 ^DE^
Nan-N (14 L/ha)		9.64 ± 0.63 ^cde^	10.10 ± 0.83 ^bc^	9.87 ± 0.90 ^BC^	9.64 ± 0.84 ^def^	10.12 ± 89 ^bcd^	9.88 ± 0.79 ^BC^
Mn-N 120 kg/ha + Nan-N 7 L/ha		9.25 ± 1.02 ^efg^	9.32 ± 0.80 ^defg^	9.29 ± 0.87 ^D^	9.10 ± 0.75 ^fg^	9.36 ± 0.70 ^efg^	9.23 ± 0.59 ^DE^
Mn-N 120 kg/ha + Nan-N 14 L/ha		10.13 ± 0.42 ^bc^	10.93 ± 0.45 ^a^	10.53 ± 0.48 ^A^	10.78 ± 0.59 ^ab^	10.87 ± 0.67 ^a^	10.83 ± 0.81 ^A^
Mn-N 240 kg/ha + Nan-N 7 L/ha		9.91 ± 0.95 ^bcd^	10.92 ± 0.79 ^a^	10.42 ± 0.97 ^A^	9.98 ± 0.81 ^cde^	10.62 ± 0.84 ^abc^	10.30 ± 0.72 ^B^
Mn-N 240 kg/ha + Nan-N 14 L/ha		9.26 ± 1.02 ^defg^	9.65 ± 0.84 ^cde^	9.46 ± 0.91 ^CD^	9.12 ± 0.89 ^fg^	9.39 ± 0.76 ^efg^	9.26 ± 0.91 ^DE^
Mean (C)		9.46 ± 0.48	9.87 ± 0.53		9.54 ± 0.79	9.81 ± 0.88	
ANOVA	df						
Cultivar (C)	1	0.164			0.087		
Fertilization Source (F)	8	0.005			0.024		
F × C	8	0.039			0.006		

Different uppercase letters indicate significant differences among the evaluated fertilization sources at *p* < 0.05, while different lowercase letters indicate significant differences among their interactions.

**Table 12 molecules-27-05640-t012:** Impact of nano- and mineral nitrogen fertilizers on the sodium concentrations (Conc; mg/L) of two wheat cultivars during the 2019/2020 and 2020/2021 seasons.

Fertilization Source (F)		First Season		Mean (F)	Second Season		Mean (F)
		Cultivar (C)			Cultivar (C)		
		Gemaiza-11	Misr-1		Gemaiza-11	Misr-1	
Untreated-control (without N)		6.41 ± 0.23 ^a^	6.14 ± 0.28 ^abcd^	6.28 ± 24 ^A^	7.27 ± 0.31 ^a^	6.29 ± 0.42 ^b^	6.78 ± 0.57 ^A^
Mn-N (120 kg/ha)		6.09 ± 0.51 ^abcd^	5.78 ± 0.24 ^def^	5.94 ± 0.31 ^ABCD^	5.80 ± 0.52 ^bcde^	5.67 ± 0.56 ^cde^	5.74 ± 0.49 ^BCD^
Mn-N (240 kg/ha)		5.90 ± 0.29 ^cde^	5.60 ± 0.28 ^efg^	5.75 ± 0.28 ^CD^	5.69 ± 0.37 ^cde^	5.63 ± 0.40 ^cde^	5.66 ± 0.79 ^D^
Nan-N (7 L/ha)		6.38 ± 0.31 ^ab^	5.87 ± 0.41 ^de^	6.13 ± 0.31 ^AB^	5.87 ± 0.61 ^bcde^	6.16 ± 0.51 ^bc^	6.02 ± 0.64 ^BC^
Nan-N (14 L/ha)		5.99 ± 0.24 ^bcde^	5.73 ± 0.30 ^defg^	5.86 ± 0.36 ^BCD^	5.79 ± 0.55 ^bcde^	5.64 ± 0.62 ^cde^	5.72 ± 0.72 ^CD^
Mn-N 120 kg/ha + Nan-N 7 L/ha		6.32 ± 0.64 ^ab^	5.85 ± 0.51 ^de^	6.09 ± 0.40 ^ABC^	6.11 ± 0.71 ^bcd^	6.02 ± 0.44 ^bcde^	6.07 ± 0.67 ^B^
Mn-N 120 kg/ha + Nan-N 14 L/ha		5.40 ± 0.30 ^fg^	5.36 ± 0.35 ^g^	5.38 ± 0.31 ^E^	5.67 ± 0.62 ^cde^	5.47 ± 0.56 ^e^	5.57 ± 0.58 ^D^
Mn-N 240 kg/ha + Nan-N 7 L/ha		5.86 ± 0.38 ^de^	5.53 ± 0.28 ^efg^	5.70 ± 0.22 ^DE^	5.67 ± 0.49 ^cde^	5.54 ± 0.67 ^de^	5.61 ± 0.69 ^D^
Mn-N 240 kg/ha + Nan-N 14 L/ha		6.30 ± 0.41 ^abc^	5.80 ± 0.30 ^def^	6.05 ± 0.24 ^ABCD^	5.88 ± 0.51 ^bcde^	5.68 ± 0.44 ^cde^	5.78 ± 0.50 ^BCD^
Mean (C)		6.07 ± 0.29	5.74 ± 0.35		5.97 ± 0.48	5.79 ± 0.61	
ANOVA	df						
Cultivar (C)	1	0.249			0.842		
Fertilization Source (F)	8	0.025			0.001		
F × C	8	0.017			0.008		

Different uppercase letters indicate significant differences among the evaluated fertilization sources at *p* < 0.05, while different lowercase letters indicate significant differences among their interactions.

## Data Availability

Not applicable.

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
