# Peer review of "Impact of Green Chitosan Nanoparticles Fabricated from Shrimp Processing Waste as a Source of Nano Nitrogen Fertilizers on the Yield Quantity and Quality of Wheat (Triticum aestivum L.) Cultivars"

_molecules, 2022, doi:10.3390/molecules27175640_

Round 1

Reviewer 1 Report

Authors need the revision of the manuscript for publishing the work for publication in Molecules journal. Some questions and suggestions are followed; 

1) The number of keywords of this article is too many. We recommend that authors should reduce the number of keywords.  

2) The length of this article is too long. We recommend that authors should move some Figures and Table to supplementary parts and reorganize some Figures and Tables. 

3) The form of references described in References part does not match with the guideline of the Molecules journal. The authors should revise references’ form accurately.

Author Response

Reviewer 1 comments

Comments and Suggestions for Authors

Authors need the revision of the manuscript for publishing the work for publication in Molecules journal. Some questions and suggestions are followed; 

Response: Thanks for the reviewer for his valuable comments and suggestions. All Comments and Suggestions have been addressed accordingly.

1) The number of keywords of this article is too many. We recommend that authors should reduce the number of keywords.  

 Response: The keywords number were reduced accordingly

2) The length of this article is too long. We recommend that authors should move some Figures and Table to supplementary parts and reorganize some Figures and Tables. 

 Response: Thanks for this suggestion. We moved some tables and figures to supplementary, additionally reorganizing the figures and the article length was reduced from 26 pages to 20 pages

3) The form of references described in References part does not match with the guideline of the Molecules journal. The authors should revise references’ form accurately.

Response: The references were reformulated following the journal guidelines

Reviewer 2 Report

Dear Author,

Thanks for your manuscript titled" Impact of Green Chitosan Nanoparticles Fabricated from Shrimp Processing Waste as a Source of Nano Nitrogen Fertilizers on the Yield Quantity and Quality of Wheat (Triticum aestivum L.) Cultivars'' I read this article with great interest, it has written well in general, however, I felt it needs some modification before consideration of this reputed journal. Below see my comments and suggestions

Comments and suggestions

1. Abstract should be more concise according to your study.

2. Please write the specific aim of your study at end of the introduction.

3. Table 1 what does mean by the first season? that is two times repeated. Please check and confirm.

4. Materials and methods section; please carefully check all the chemicals, reagent, and the company name has been written correctly.

5. Figures 1 and 2 should provide a real FTIR image.

6. In the discussion section please indicate your advantages and limitation of this study.

7. In Conclusion summarize the main points of your research.

8. Lot of typos and grammatical errors throughout the manuscript that should be checked correctly before submission.

Author Response

Reviewer 2#

Comments and Suggestions for Authors

Dear Author,

Thanks for your manuscript titled" Impact of Green Chitosan Nanoparticles Fabricated from Shrimp Processing Waste as a Source of Nano Nitrogen Fertilizers on the Yield Quantity and Quality of Wheat (Triticum aestivum L.) Cultivars'' I read this article with great interest, it has written well in general, however, I felt it needs some modification before consideration of this reputed journal. Below see my comments and suggestions

 Response: Thanks for the reviewer for his valuable comments and suggestions. All Comments and Suggestions have been addressed accordingly.

Comments and suggestions

  1. Abstract should be more concise according to your study.

Response: Thanks for the reviewer for this suggestion. The abstract was reformulated accordingly.

  1. Please write the specific aim of your study at end of the introduction.

Response: Thanks for the reviewer for this comment. The aim was defined at the end of introduction.

  1. Table 1 what does mean by the first season? that is two times repeated. Please check and confirm.

Response: Acutely, the study was conducted at two seasons, so we analyzed the tested soils at two seasons.

  1. Materials and methods section; please carefully check all the chemicals, reagent, and the company name has been written correctly.

Response: Done accordingly.

  1. Figures 1 and 2 should provide a real FTIR image.

Response: Done accordingly, the real images were cleared in Figure 1 C and D

  1. In the discussion section please indicate your advantages and limitation of this study.

Response: Done accordingly.

  1. In Conclusion summarize the main points of your research.

Response: Done accordingly.

  1. Lot of typos and grammatical errors throughout the manuscript that should be checked correctly before submission.

Response: The linguistic errors were checked by expert.